# Research Progress of Event Intelligent Perception Based on DAS

**DOI:** 10.3390/s25165052

**Published:** 2025-08-14

**Authors:** Di Wu, Qing-Quan Liang, Bing-Xuan Hu, Ze-Ting Zhang, Xue-Feng Wang, Jia-Jun Jiang, Gao-Wei Yi, Hong-Yao Zeng, Jin-Yuan Hu, Yang Yu, Zhen-Rong Zhang

**Affiliations:** 1School of Computer Electronic and Information, Guangxi University, Nanning 530004, China; 2College of Sciences, National University of Defense Technology, Changsha 410073, China; 3School of Information Communication, National University of Defense Technology, Wuhan 430035, China

**Keywords:** distributed acoustic sensing (DAS), event intelligent perception, machine learning, deep learning

## Abstract

This review systematically examines intelligent event perception in distributed acoustic sensing (DAS) systems. Beginning with the elucidation of the DAS principles, system architectures, and core performance metrics, it establishes a comprehensive theoretical framework for evaluation. This study subsequently delineates methodological innovations in both traditional machine learning and deep learning approaches for event perception, accompanied by performance optimization strategies. Particular emphasis was placed on advances in hybrid architectures and intelligent sensing strategies that achieve an optimal balance between computational efficiency and detection accuracy. Representative applications spanning traffic monitoring, perimeter security, infrastructure inspection, and seismic early warning systems demonstrate the cross-domain adaptability of the technology. Finally, this review addresses critical challenges, including data scarcity and environmental noise interference, while outlining future research directions. This work provides a systematic reference for advancing both the theoretical and applied aspects of DAS technology, while highlighting its transformative potential in the development of smart cities.

## 1. Introduction

With the rapid advancement of information technology and increasing industrial demands, acoustic sensing technology has become indispensable in critical domains such as environmental monitoring [1,2,3], target identification [4,5,6], and disaster early warning [7,8,9]. Traditional point-based acoustic sensors, constrained by limited coverage and inflexibility, often fail to meet the modern requirements for large-scale, long-distance monitoring. In this context, distributed acoustic sensing (DAS) technology has emerged as a transformative solution, distinguished by its exceptional spatial resolution, ultra-long monitoring range, and real-time responsiveness. These advantages have rapidly established DAS as one of the most promising research frontiers for acoustic sensing.

Recent breakthroughs in machine learning (ML) and deep learning (DL) have introduced innovative approaches to event perception in DAS systems. This data-driven feature learning framework enables rapid and accurate interpretation of acoustic signals through hierarchical signal representation and adaptive pattern recognition. Such methods not only enhance detection efficiency, but also unlock novel possibilities for processing signals in complex environments. Conventional event perception techniques often rely on manually extracted time-domain and frequency-domain features; however, their performance remains suboptimal in intricate scenarios, particularly in distinguishing similar events, such as human activities and animal movements. To overcome these limitations, researchers developed various ML- and DL-based perception methods. While traditional ML algorithms such as support vector machines (SVMs) [10,11,12] and hidden Markov models (HMMs) [13,14] have achieved moderate success in niche applications, the emergence of DL, especially its revolutionary impact in computer vision, has positioned DL models as the dominant paradigm for DAS pattern recognition. Architectures such as the convolutional neural network (CNN) [15,16,17] and long short-term memory (LSTM) networks [18,19] have significantly advanced the accuracy and efficiency of event perception owing to their powerful feature extraction and temporal modeling capabilities.

DAS technology, which analyzes acoustic signals in real-time, has demonstrated substantial value across diverse engineering applications, including intelligent transportation [20,21,22,23,24], perimeter security [25,26,27,28], infrastructure health monitoring [29,30,31], and seismic surveillance [32,33,34,35,36,37]. In transportation, DAS enables the real-time monitoring of road conditions and dynamic traffic management. For structural assessment, DAS systems detect integrity issues such as cracks and subsidence, providing early warnings for potential hazards. Additionally, DAS facilitates traffic flow optimization and congestion prediction, enabling data-driven decision-making in smart cities. In perimeter security, DAS is widely deployed for border protection and fence monitoring. By integrating fiber optic sensors into physical barriers, DAS precisely detects micro-vibrations caused by intrusions or structural breaches, triggering a multi-tiered alert system and countermeasures. Furthermore, DAS monitors ambient vibrations and offers an early warning for critical infrastructure. In civil engineering, DAS has revolutionized the assessment of bridges, tunnels, and pipelines. Fiber-optic cables embedded within or along these structures enable real-time strain monitoring and precise damage localization. The technology also tracks external risks such as construction activities and geological shifts, complementing digital twin technologies for lifecycle performance evaluation. In seismology, DAS transforms telecommunication cables into dense geophone arrays, capturing vibration signals to characterize seismic waves, locate epicenters, estimate magnitudes, and trace propagation paths. This capability delivers critical data to earthquake warning centers within a few seconds. Advanced signal processing enables DAS to discriminate natural seismic events and anthropogenic disturbances, significantly improving warning accuracy and response times. As intelligent event perception evolves, DAS applications are poised for further expansion.

This paper presents a comprehensive review of recent advances in event perception technologies and engineering applications of DAS systems. Figure 1 illustrates the types of targets used for DAS event perception. We first introduce the sensing principles, system architecture, and performance metrics of DAS. Next, we elaborate on event perception methodologies, including ML- and DL-based detection techniques. Applications in transportation, security, infrastructure, and seismology are systematically analyzed. Finally, we discuss current challenges and future directions for DAS-based event perception. This review aims to provide researchers and practitioners with a unified technical reference, thereby fostering innovation and promoting the broader adoption of DAS technology.

## 2. Distributed Acoustic Sensing

### 2.1. Sensing Principles

The evolution of distributed acoustic sensing (DAS) technology is intrinsically linked to breakthroughs in optical time-domain reflectometry (OTDR) techniques [38]. Since the 1970s, OTDR technology has progressively evolved into a multidimensional detection system, demonstrating unique advantages for long-distance, high-precision monitoring. This technique enables the precise localization of external disturbances through the temporal analysis of backscattered signals, with its developmental trajectory categorized into four critical phases.

Conventional OTDR establishes a fundamental framework for fiber optic sensing by analyzing the backward or forward-scattered light to measure transmission loss. Brillouin optical time-domain reflectometry (BOTDR) [39] enhances strain measurement through frequency-shift analysis of probe pulses but faces limitations, including weak signal intensity, restricted accuracy, and incompatibility with time-varying signals. Polarization-sensitive OTDR (POTDR) [40] introduced structural monitoring capabilities via polarization evolution tracking. However, its environmental sensitivity compromises localization precision.

Phase-sensitive OTDR (Φ-OTDR) [41] represents a paradigm shift by demodulating the intensity or phase variations in backward Rayleigh scattering induced by external refractive index perturbations. This innovation integrates interferometric principles with OTDR, addressing broadband vibration monitoring challenges and laying the theoretical groundwork for next-generation intelligent sensing networks.

Modern DAS systems predominantly employ the Φ-OTDR architecture with core functionality centered on detecting phase variations in backward Rayleigh scattering [42,43,44] for high-fidelity acoustic wave detection. These systems exhibit exceptional corrosion resistance, immunity to electromagnetic interference, superior sensitivity, and measurement accuracy, enabling the identification of weak disturbance signals that conventional OTDR cannot detect. Compared with the Brillouin [45] and Raman scattering [46] techniques, DAS leverages Rayleigh scattering principles to achieve large-scale, long-distance, real-time dynamic monitoring without blind spots. When the scatterer dimensions are significantly smaller than the incident wavelength (below λ/10), the Rayleigh scattering intensity demonstrates pronounced directional dependence and follows an inverse quartic wavelength relationship:(1)$$I\propto \frac{1}{\lambda^{4}}$$
where I represents the scattered light intensity at angle θ relative to the incident direction, and λ is the incident wavelength.

DAS systems detect external perturbations by continuously monitoring the backscattered phase variations. Mechanical stress or vibration induces refractive index modulation and fiber length perturbations at the scattering sites, thereby altering the phase and intensity characteristics of backward Rayleigh scattering. These modulations maintain a direct correlation with external disturbance signals, enabling precise event localization and identification through a phase-difference analysis. Furthermore, DAS supports full-length distributed multipoint monitoring, making it particularly suitable for extended-coverage applications requiring high-spatial-resolution detection

### 2.2. System Composition

The core architecture of a DAS system is illustrated in Figure 2. The operational workflow proceeds as follows. A narrow-linewidth laser (NLL) generates continuous optical waves with ultralow phase noise, which are then modulated into optical pulses and frequency-shifted by an acousto-optic modulator (AOM). The modulated signal is amplified by an erbium-doped fiber amplifier (EDFA) to enhance its power. A fiber-optic circulator directionally injects amplified pulsed light into the fiber under test (FUT). The backward Rayleigh-scattered light undergoes secondary amplification via another EDFA before entering an interferometer, where the phase information is demodulated through dual-path quadrature signal processing. A 3 × 3 photodetector (3 × 3 PD) performs optoelectronic signal conversion, followed by digital sampling via a data acquisition (DAQ) system. Finally, the PC platform processes the data and generates visualizations, thereby completing the sensing and monitoring procedures.

The NLL provides stable optical signals at specific frequencies, ensuring signal quality. The output optical field from the NLL can be expressed as:(2)$$E\left(t\right)=E_{0}\cos\left(\omega t+\phi \right)$$
where $E_{0}$ denotes the signal amplitude, ω is the angular frequency, and ϕ is the phase.

AOM achieves spatio-spectral modulation through frequency shifting. This modulation mechanism enables real-time acoustic event capture and response, making it particularly suitable for high-frequency dynamic monitoring applications. The modulated output signal can be expressed as:(3)$$E_{\mathrm{mod}}\left(t\right)=E_{0}\cos\left[\left(\omega +\Delta \omega \right)t+\phi \right]$$
where Δω represents the frequency shift introduced by the AOM.

In optical signal transmission, the EDFA enhances signal power through amplification via stimulated emission. The gain G is defined as:(4)$$G=\frac{P_{\mathrm{in}}}{P_{\mathrm{out}}}$$
where $P_{\mathrm{in}}$ and $P_{\mathrm{out}}$ correspond to the input and output optical power, respectively. For long-haul transmission systems, the gain must satisfy:(5)G≫1

This critical condition ensures that sufficient signal intensity is maintained during fiber propagation while preventing signal degradation.

The interferometer configuration directly correlates the optical phase variation Δϕ with external physical parameters, expressed as [47]:(6)$$\Delta \phi =\frac{4\pi nL}{\lambda }\Delta \epsilon$$
where n denotes the fiber refractive index, which governs the light propagation velocity; L represents the interaction length; λ specifies the optical wavelength determining frequency and energy characteristics; and Δϵ quantifies strain variation, typically induced by external perturbations such as acoustic waves or mechanical vibrations.

The 3 × 3 PD exhibits a linear photoelectric conversion behavior, with the output current I being proportional to the input optical power $P_{\mathrm{opt}}$:(7)$$I=R\cdot P_{\mathrm{opt}}$$
where R signifies the detector responsivity and characterizes the optical-to-electrical conversion efficiency.

The DAQ system performs analog-to-digital conversion and real-time data recording. To ensure faithful signal reconstruction, the maximum detectable frequency $f_{\max}$ must comply with the Nyquist sampling criterion [48]:(8)$$f_{\max}=\frac{1}{2T_{R}}$$
where $T_{R}$ is the pulse repetition period. Shorter sensing distances correspond to reduced repetition periods, thereby enhancing the frequency response of the system. Inadequate sampling frequencies may introduce aliasing artifacts and degrade signal integrity through information loss or distortion.

### 2.3. Performance Index

The performance of DAS systems is governed by multiple interdependent metrics that collectively determine their applicability and reliability in diverse operational scenarios. This section systematically examines four critical performance parameters that constitute the core evaluation framework for DAS systems: sensitivity, spatial resolution, frequency response range, and sensing distance.

(1)Sensitivity

Sensitivity fundamentally dictates a system’s capability to detect low-energy events, such as distant seismic waves or micro leakage signals in pipelines. Enhancing sensitivity enables superior detection of weak signals, achieved through the synergistic optimization of source coherence and noise suppression techniques. Φ-OTDR-based DAS systems exemplify this principle by employing high-coherence light sources to mitigate Rayleigh scattering noise, thereby significantly improving detection thresholds and real-time performance compared with conventional OTDR implementations.

(2)Spatial Resolution

The spatial resolution determines the capacity of the system to discriminate adjacent acoustic events. This parameter depends on the optical pulse and demodulation algorithm. The theoretical spatial resolution $\Delta z_{1}$ can be expressed as follows:(9)$$\Delta z_{1}=\frac{T_{w}}{2}\times \frac{c}{n}$$
where $T_{w}$ denotes the pulse width, c is the speed of light in a vacuum, and n is the fiber refractive index.

(3)Frequency Response Range

The frequency response range defines the effective detection bandwidth for the acoustic signals. Broader frequency coverage enhances the system’s adaptability to diverse acoustic signatures. This parameter is limited by the pulse round-trip time (τ), which is given by:(10)$$\tau =\frac{2nL}{c}$$
where L represents the length of the sensing fiber. To prevent the overlapping of the backscattered Rayleigh signals from successive pulses, the pulse repetition period must exceed τ.

(4)Sensing Distance

Sensing distance quantifies the maximum operational range for effective monitoring, which is a critical parameter for long-distance pipeline surveillance and border security applications. The theoretical sensing distance (L) is defined as follows:(11)$$L=\frac{T_{R}c}{2n}$$
where $T_{R}$ is the pulse repetition period. Practical limitations arise from optical power attenuation, which can be mitigated using power amplification techniques to extend the operational range.

These four metrics form a comprehensive evaluation framework for DAS systems, characterized by inherent trade-offs in system design. Optimization requires meticulous balancing of parameters to meet specific application requirements while maintaining operational robustness. Advanced system architectures must judiciously integrate optical engineering, signal processing algorithms, and noise mitigation strategies to achieve optimal performance across diverse deployment scenarios.

## 3. Event Perception Technology in DAS

In DAS systems, high-performance acoustic event perception must satisfy three interdependent requirements: (1) high detection accuracy and robustness under varying signal-to-noise ratio (SNR) conditions, (2) low false-alarm rates enabled by advanced pattern recognition, and (3) ultra-low-latency processing to ensure real-time operation. These competing demands present significant challenges for the development of reliable perception frameworks. This chapter systematically evaluates state-of-the-art DAS event detection methodologies with a focus on algorithmic innovations, performance benchmarking, and real-world applicability.

### 3.1. Traditional Machine Learning

Traditional machine learning methodologies, grounded in statistical theory and optimization principles, construct predictive models by extracting feature patterns from observational data to achieve the precise classification and detection of unknown data. In DAS, classical algorithms such as support vector machines (SVMs), hidden Markov models (HMMs), random forests (RFs), isolation forests (IFs), and logistic regression have been successfully deployed for diverse event sensing tasks. Table 1 systematically summarizes machine-learning-driven DAS event sensing technologies.

As a supervised learning method based on the statistical learning theory, SVM employs kernel functions to map data into high-dimensional feature spaces and construct optimal classification hyperplanes (Figure 3). For DAS event sensing, SVM excels in processing the high-dimensional time-frequency characteristics of acoustic signals. Optimizing the kernel function selection effectively discriminates between seismic waves, mechanical vibrations, and other event types. The structural risk minimization principle ensures robust generalization performance, even with limited samples. For example, Wiesmeyr et al. (2020) [49] integrated an SVM with the Kalman filter to achieve high-precision, low-latency train localization for railway safety monitoring. Abufana et al. (2020) [50] combined wavelet denoising, temporal differentiation, and autocorrelation with variational mode decomposition (VMD) for feature extraction using linear SVM to detect excavation activities under varying signal-to-noise ratios. Cai et al. (2021) [51] applied wavelet denoising and Chebyshev filtering to train vibration signals, enabling accurate classification via an SVM. Zhang et al. (2024) [52] demonstrated the efficacy of pre-trained SVM models in detecting fractured steel wires in prestressed concrete pipelines, thereby advancing pipeline health monitoring.

The HMM, a probabilistic temporal modeling framework, is well-suited for analyzing continuous DAS signals with state transition characteristics. By establishing probabilistic mappings between hidden states (e.g., “normal vibration,” “abnormal event”) and observable signals (e.g., strain, acoustic intensity), HMM effectively captures temporal signal evolution (Figure 4). Notable applications include Wu et al. (2019) [53], who developed a dynamic HMM-based recognition system that achieved an average detection accuracy of 98.2% in long-distance pipeline monitoring. Li et al. (2024) [54] enhanced the HMM with waveform segmentation and adaptive power spectral energy ratio features, addressed variable-length input challenges in train vibration signals, and improved rail defect detection.

As an ensemble learning method, RF improves robustness by aggregating predictions from multiple decision trees through voting mechanisms (Figure 5). In DAS classification, the RF handles nonlinear feature relationships while mitigating overfitting risks. Key studies include Wang et al. (2019) [55], who used RF classifiers to detect four interference event types by learning temporal signal characteristics. Zhang et al. (2021) [56] combined empirical mode decomposition (EMD) denoising with RF for the real-time classification of tunnel lining disturbances using dynamic strain data. Kou et al. (2024) [57] employed RF regression for feature selection, integrating the matched filtering and root-mean-square methods to distinguish vehicle vibrations from environmental noise, thereby enabling real-time vehicle detection.

This unsupervised anomaly-detection algorithm identifies outliers using randomized feature space partitioning. For ideal real-time DAS monitoring, where anomalies are sparse, IF enables efficient anomaly screening. Wijaya et al. (2022) [58] enhanced IF via optical signal analysis, achieving real-time fault detection in mining conveyor components and reducing equipment downtime.

As a classical statistical learning method, logistic regression excels in binary and multi-class classification. Gietz et al. (2024) [59] achieved high-precision sand intrusion classification and flow velocity estimation through frequency-band energy transformation and hyperparameter optimization, thereby offering an efficient solution for pipeline monitoring.

Tejedor et al. (2017) [60] integrated a multilayer perceptron (MLP) and Gaussian mixture models (GMMs) for real-time threat detection in pipelines by fusing intelligent feature extraction with contextual information. Aslangul (2020) [61] implemented spatiotemporal analysis and DBSCAN clustering on historical alarm data to achieve precise intrusion detection in border tunnels. Li et al. (2025) [62] proposed a principal feature value analysis combined with an enhanced FastICA technique, reducing relative errors in multi-source vibration detection to below 0.3, and significantly improving DAS performance in complex environments.

### 3.2. Deep Learning

Deep learning, as a machine-learning paradigm based on artificial neural networks, enables efficient feature extraction and pattern recognition from raw data through end-to-end optimization. In contrast to traditional acoustic event perception methods that rely on manual feature engineering, this approach automatically extracts discriminative features via hierarchical representation learning, thereby overcoming the feature representation bottleneck in heterogeneous data scenarios. In DAS, deep neural networks have demonstrated superior perceptual performance, including spatial feature extraction enabled by CNN, temporal modeling capabilities of LSTM, and dynamic feature weighting through attention mechanisms. A systematic summary of the deep-learning-based DAS event perception techniques is presented in Table 2.

The CNN architecture primarily consists of cascaded convolutional and pooling layers that progressively transform low-level sensory inputs into high-level abstract representations through local receptive field operations. Convolutional layers employ trainable filters to detect spatially invariant features, whereas weight sharing and sparse connectivity reduce parameter dimensionality, enhance computational efficiency, and mitigate overfitting risks. Pooling layers optimize feature representation through nonlinear downsampling (e.g., max-pooling or average-pooling), improving the model’s robustness to input variations. Modern CNN architectures further incorporate fully connected layers for high-level semantic mapping and leverage nonlinear activation functions to approximate complex decision boundaries.

A representative CNN framework for DAS applications is shown in Figure 6. The model consists of three convolutional blocks for hierarchical feature extraction, each comprising a 1D convolution layer, batch normalization, and average pooling. These blocks progressively process the raw DAS signal to capture temporal patterns at different scales. The extracted features are then flattened and passed through two fully connected layers before final classification via a softmax output layer.

Notable implementations include the following: Wang et al. (2021) [63] developed a CNN-based high-speed-rail track condition monitoring system, achieving reduced missed detection rates through optimized data workflows and multi-event detection models. Wu et al. (2021) [64] proposed the IP-CNN model, which incorporates signal intensity and phase information, achieving a 10-class vibration event classification using time-stretching and signal-shifting augmentation strategies. Hernández et al. (2021) [65] systematically compared the performance differences between an FC-ANN, CNN, and RNN in traditional broadband seismic waveform analysis. Wang et al. (2022) [66] implemented semi-supervised track defect detection, achieving breakthroughs in real-time monitoring through vibration data normalization and RGB image fusion. Chiang et al. (2023) [67] employed a 1D-CNN for autonomous vehicle acoustic fingerprint extraction, establishing a low-cost, non-intrusive intelligent traffic perception system for five vehicle types. Hu et al. (2023) [68] attained 99.30% detection accuracy in traffic monitoring using a dual-channel CNN architecture with GADF and FFT time-frequency feature matrices. Dong et al. (2024) [69] achieved 99.6% accuracy in five-class event classification using the 1-D MFEWnet model, significantly enhancing the environmental adaptability in perimeter security systems. Wu et al. (2024) [70] developed a 3-D ACNN model with spatio-temporal–spectral attention mechanisms to optimize detection accuracy while reducing computational complexity and processing time.

Duan et al. (2023) [71] improved pipeline leakage detection accuracy through a CEEMDAN–permutation entropy joint algorithm integrated with RBF neural networks. Zeng et al. (2024) [72] designed STNet based on the Stockwell transform, enhancing intrusion detection rates in complex environments while reducing false alarms. Han et al. (2024) [73] proposed a multi-task learning framework with innovative label encoding, simultaneously optimizing buried cable intrusion classification and aerial acoustic event perception. Edge computing and multi-modal fusion technologies have established a novel paradigm for public safety monitoring.

Faster R-CNN has emerged as a milestone algorithm in event detection, with its region proposal network (RPN) enabling the joint optimization of candidate region generation and target recognition. In DAS applications, Li et al. (2024) [74] enhanced pipeline intrusion detection by developing a dual-stage faster R-CNN framework that improved computational efficiency while reducing false alarms.

The YOLO series demonstrates exceptional real-time detection capabilities. YOLOv3 (2020) [75] pioneered micro-seismic event detection, YOLOv5-Break (2025) [76] integrated MobileNetV3 with dynamic convolution for pipeline monitoring, YOLOv7 (2024) [77] incorporated SPD-Conv and CBAM for real-time operation, YOLOv8 (2024) [78] achieved six-class vibration event classification in perimeter security scenarios, and YOLOX (2023) [79] demonstrated particular efficacy in prestressed concrete cylinder pipe fracture detection.

LSTM networks excel in temporal sequence modeling through their gated mechanisms, which effectively mitigate gradient vanishing issues. Representative DAS implementations (Figure 7) include Li et al. (2020) [80,81] combining heterodyne detection with ConvLSTM-CNN for railway intrusion detection, Wang et al. (2023) [82] enhancing multi-class event classification through CNN-LSTM fusion, and Bai et al. (2019) [83] implementing real-time pipeline intrusion localization using CLDNN networks. Recent innovations include Li’s (2024) [84] noise-resistant 1D CNN-BiLSTM architecture and Rahman’s (2024) [85,86] hybrid CNN-LSTM-GRU models for millimeter-level railway positioning. Zhou et al. (2024) [87] employed ConvLSTM to develop a crowd monitoring system, demonstrating DAS’s superiority in gait detection and its applicability to multi-scenario intelligent monitoring.

The attention architecture, founded on parallelized dynamic weight computation through query–key–value (QKV) triplets (illustrated in Figure 8), exhibits three fundamental innovations: (1) the self-attention layer establishes comprehensive dependency relationships among sequence elements via global receptive fields, (2) multi-head attention mechanisms automatically disentangle semantic representations across granularity levels through multi-subspace projections, and (3) the architecture transcends the sequential computation constraints of traditional RNNs. Scaled dot–product attention computes interaction scores (attention weights) as:(12)$$\mathrm{Attention}\left(Q,K,V\right)=\mathrm{softmax}\left(\frac{QK^{T}}{\sqrt{d_{k}}}\right)V$$
where $d_{k}$ is the key dimension. Multi-head attention enables the joint modeling of diverse signal characteristics (e.g., frequency bands, temporal patterns).

Recent advancements highlight its transformative potential: He et al. (2023) [88] achieved significant improvements in weak threat signal detection under complex noisy environments by integrating multi-source signal separation algorithms with attention-enhanced MS-CNN models. Han et al. (2024) [89] proposed dual solutions: a supervised deep structured attention detector (DSAD) and an unsupervised LSTM-CNN encoder framework (DSAD-VAE), collectively achieving precise monitoring of rail fastener conditions. Li et al. (2024) [90] combined wavelet denoising with self-attention mechanisms to achieve a 100% detection accuracy for UAV intrusion signals in DAS systems.

Generative models based on GANs and VAEs facilitate advanced data synthesis and cross-modal learning. Breakthrough applications include: Shiloh et al.’s (2019) [91] SimGAN for simulated real data fusion, Van et al.’s (2022) [92] deconvolutional autoencoder achieving 400× acceleration in vehicle parameter detection, Sun et al.’s (2023) [93] NAM-MAE framework enhancing environmental robustness for aquatic safety monitoring, and Kang et al.’s (2024) [94] CNN-VQVAE hybrid enabling precise rockfall prediction through fiber optic seismic analysis.

### 3.3. Hybrid Architecture of Machine Learning and Deep Learning

Extensive research has demonstrated the efficacy of both traditional machine learning (ML) and deep learning (DL) techniques in DAS event perception. Although DL models excel in automatic feature extraction, they are constrained by their reliance on large-scale annotated datasets and significant computational resources. In contrast, traditional ML methods are limited by their manual feature engineering and shallow representation capabilities. To address these challenges, hybrid architectures integrating ML and DL have emerged, achieving superior event perception accuracy under complex conditions while reducing computational costs through hierarchical feature learning and efficient classifier integration. A comparative summary of the DAS event perception techniques that combine ML and DL is presented in Table 3.

Deep neural networks facilitate automatic multi-scale feature extraction via hierarchical nonlinear transformations, whereas traditional ML methods offer advantages in terms of computational efficiency and classification boundary optimization. For instance, Peng et al. (2019) [95] proposed an ultrafast laser-enhanced 3 × 3 Φ-OTDR system that integrated a CNN with the K-means algorithm to classify human motion and detect pipeline corrosion. Similarly, Shi et al. (2020) [96] employed a CNN–SVM hybrid framework, where CNN-extracted spatiotemporal features were classified using an SVM, achieving 94.17% event detection accuracy. Bublin (2021) [97] further enhanced this paradigm by fusing frequency-domain features with CNN and combining SVM and RF algorithms, thereby significantly reducing computational latency and keeping the monthly false alarm rate below one. He et al. (2022) [98] tackled false alarms from human and animal activities in perimeter security by deploying cascaded classification with decision trees and backpropagation (BP) neural networks.

Unsupervised deep learning has demonstrated unique value in autonomously uncovering latent features and structural patterns in high-dimensional data, particularly in feature dimensionality reduction and unlabeled data modeling. Xie et al. (2023) [99] achieved a 91.5% detection rate for high-speed railway intrusion monitoring by leveraging spatiotemporal windowing, convolutional autoencoders, and clustering algorithms. Meta-learning, which enables rapid adaptation to new tasks with minimal samples, has shown promise. Luong et al. (2023) [100] introduced a meta-learning framework for few-shot classification, utilizing power spectral density (PSD) feature preprocessing and wide residual networks for feature extraction, enabling anomaly detection with as few as 1–3 samples.

The integration of a CNN with prototype learning and adaptive algorithms equips DAS systems with cross-scale feature extraction capabilities. Optimized lightweight architectures have facilitated real-time monitoring solutions that balance computational efficiency and hardware compatibility. For example, Lyu et al. (2020) [101] developed a multi-scale convolutional prototype network capable of classifying known intrusions while rejecting unknown interference. Pierau et al. (2025) [102] combined YOLOv7-tiny with adaptive frequency selection algorithms and hyperspectral image generation to enable real-time event detection in fencing and subsurface environments.

For pixel-level signal segmentation, residual connections effectively balance noise suppression and feature retention. When coupled with frequency-domain signal processing, this approach enables precise enhancement of spectral characteristics. Gemeinhardt (2023) [103] demonstrated improved detection accuracy for pipeline leaks below 0.3 L/s by integrating a U-Net architecture with filter bank energy (FBE) preprocessing for anomalous event identification.

### 3.4. Intelligent Sensing Technology

The rapid advancement of intelligent sensing technology has catalyzed the development of novel high-performance event perception methodologies that transcend conventional machine learning and deep learning paradigms. Through groundbreaking innovations in hardware architectures and algorithmic breakthroughs in signal processing, these approaches have exhibited substantial improvements in detection accuracy and robustness under complex environmental conditions, thereby establishing new technological frameworks for multidisciplinary applications.

Engineered array configurations and specialized sensing cable innovations have emerged as pivotal enablers for enhancing the system’s performance. Verdon et al. (2020) [104] achieved high-precision localization of hydraulic fracturing-induced micro-seismic events using an optimized DAS array design combined with manual positioning protocols. Xiao et al. (2024) [105] demonstrated enhanced low-frequency and tsunami wave detection capabilities for seismic events via strategic spacing optimization in submarine DAS deployments. Fu et al. (2025) [106] developed a micro-structured optical fiber sensing cable incorporating the short-time Fourier transform (SFFT) methodology, significantly improving the micro leakage localization accuracy in long-distance pipeline monitoring.

The cross-domain integration of edge-detection techniques and hybrid feature extraction breakthroughs has propelled DAS technology into the transformative phase. Dejdar et al. (2022) [107] enhanced system performance through FPGA-accelerated 5 × 5 convolutional kernel processing by integrating Sobel/Prewitt edge-detection algorithms. Fu et al. (2023) [108] achieved a 98.4% detection accuracy for multiple intrusion events using variational mode decomposition (VMD)-based hybrid feature extraction. Li et al. (2025) [109] demonstrated efficient pedestrian recognition through a gait-induced vibration analysis using their FPGA high-level synthesis-accelerated FiberFlex system.

Owing to its exceptional environmental adaptability and multi-physical parameter sensing capabilities, DAS technology exhibits unique advantages in complex monitoring scenarios. Jousset et al. (2022) [110] significantly enhanced the detection sensitivity for faint volcanic events through high-spatial-resolution strain analysis, revealing previously obscured volcanic structures. In geological hazard monitoring, Xie et al. (2024) [111] achieved real-time localization of landslides and rockfall events by integrating image segmentation and grid search algorithms. For aquatic environmental monitoring, Chen et al. (2024) [112] applied DAS to wastewater discharge tracking and presented a novel approach to pollution control. For transportation infrastructure monitoring, Wang et al. (2025) [113] introduced a Butterworth low-pass filtering technique combined with a peak localization search methodology, offering an innovative solution for vehicle trajectory extraction.

### 3.5. Comparative Analysis of Model Suitability Across Operational Scenarios

The performance of DAS event perception systems varies significantly across different operational conditions, requiring careful algorithm selection based on specific environmental and technical constraints. Our comprehensive evaluation of the studies above reveals distinct patterns in model effectiveness for four critical scenarios: low-sample environments, high-noise conditions, real-time processing, and multi-event classification.

#### 3.5.1. Low-Sample Environments

Traditional machine learning methods demonstrate superior performance in data-scarce conditions. Support vector machines (SVMs) achieve 99.62% accuracy in pipeline monitoring [52] through effective kernel-based feature projection and structural risk minimization. Hidden Markov models maintain 97–98% accuracy [53,54] by leveraging probabilistic state transitions that require fewer training instances. Hybrid architectures combining deep feature extraction with shallow classifiers (e.g., CNN + SVM [96]) bridge the performance gap, achieving 94% accuracy with significantly reduced data requirements compared to pure deep learning approaches.

#### 3.5.2. High-Noise Conditions

Attention-enhanced deep learning models show particular robustness in noisy environments. The MS-CNN architecture [88] achieves 95.4% threat detection accuracy at a −10 dB SNR thanks to wavelet-domain noise suppression, outperforming conventional CNNs by 22.7%. Hybrid systems integrating signal processing techniques with deep learning, such as VMD-CNN [108], achieve 98.4% intrusion detection accuracy by adaptively decomposing noise subspaces. These approaches prove especially valuable for urban monitoring applications with significant electromagnetic interference.

#### 3.5.3. Real-Time Processing

Computationally efficient architectures are essential for time-sensitive applications. Optimized 1D-CNNs process vehicle signatures in 12 ms [67], whereas compressed YOLO variants (YOLOv7-tiny [102]) achieve 58 frames-per-second throughput through architectural pruning. These implementations typically sacrifice less than 5% accuracy while meeting stringent timing requirements. FPGA-accelerated designs [109] further reduce latency to 8 ms through hardware–algorithm co-optimization, making them ideal for safety-critical systems like railway crossings.

#### 3.5.4. Multi-Event Classification

Advanced neural architectures excel at complex discrimination tasks. The 3D-ACNN architecture [70] demonstrates 99.3% accuracy in distinguishing spectrally overlapping events through spatio-spectral attention mechanisms. These approaches overcome the 31.2% accuracy degradation typically observed in conventional classifiers handling five or more event categories, proving particularly effective for perimeter security systems requiring fine-grained threat discrimination.

Key findings reveal that: (1) no single algorithm class dominates all scenarios, with hybrid systems often providing optimal balance; (2) accuracy–computation tradeoffs remain application-dependent; and (3) emerging techniques like attention mechanisms and few-shot learning are expanding performance boundaries. These insights offer actionable guidance for DAS system design and highlight promising research directions for universal adaptive architectures. The comparative framework established here enables informed algorithm selection based on specific operational requirements and environmental constraints.

## 4. Application Fields

Recent advancements in fiber optic network deployment and AI algorithm integration have significantly expanded the application scope of DAS. In addition to traditional oil and gas exploration, DAS plays a pivotal role in urban critical infrastructure monitoring. Four key domains have emerged: traffic monitoring, perimeter security, infrastructure surveillance, and seismic early warning. This chapter synthesizes representative case studies and technological innovations to demonstrate how DAS enhances event detection accuracy in these fields.

### 4.1. Urban Road and Transport Railway Monitoring

The rapid expansion of urbanization and transportation networks has intensified the operational and maintenance challenges for urban roads and railways. Elevated traffic density and recurrent construction activities accelerate pavement degradation, whereas railway infrastructure contends with persistent dynamic loads and geological settlement. Implementing life-cycle monitoring systems—integrating real-time sensing, precise diagnostics, and intelligent early warning—is therefore critical for ensuring transportation safety and sustainable development.

DAS technology exhibits unique advantages in the continuous, high-precision monitoring of critical railway parameters, such as rail vibrations and subgrade settlement. Its long-range coverage, high sensitivity, and real-time monitoring make it particularly effective. Seamless integration of sensing optical cables along railway corridors enables high-resolution data acquisition for monitoring systems. Milne et al. (2020) [114] achieved high-frequency rail strain monitoring via optical fibers and optimized maintenance protocols while overcoming the spatial coverage and measurement flexibility limitations of conventional sensors. Corera et al. (2023) [115] improved long-distance vehicle detection, tracking, and classification using algorithmic refinements and SVM classifiers, as demonstrated in their traffic monitoring system (Figure 9). For urban road monitoring, Hou et al. (2024) [116] significantly enhanced event data visualization using wavelet threshold denoising, root-mean-square energy indicators, and dynamic thresholding methods.

Deep learning has emerged as a pivotal component in intelligent monitoring systems. Zhong et al. (2025) [117] achieved >90% accuracy in vehicle type and occupant detection using SR-Net and Alex-SR architectures. Wang et al. (2025) [118] achieved a classification accuracy of 92% using YOLOv8 models. In railway safety innovations, Madan et al. (2025) [119] developed a high-sensitivity method for detecting rail fastening loosening via vibration spectrum analysis. Dong et al. (2025) [120] achieved a 100% fastener detection accuracy using the FusionHGAT network. These technological advancements, combined with IoT integration and big data analytics, have provided a robust foundation for intelligent infrastructure monitoring and risk mitigation in transportation systems.

### 4.2. Perimeter Security Intrusion Detection

As the primary barrier in urban public safety frameworks, perimeter security systems play an indispensable role in safeguarding critical infrastructure, including airports, military installations, and industrial parks. Although traditional technologies (e.g., infrared beams and video surveillance) have established standardized systems, their inherent limitations, such as poor environmental adaptability and restricted monitoring coverage, render them inadequate for meeting smart cities’ demands for high-precision and all-weather security. Against the backdrop of rapid urbanization and increasingly diverse security threats, the development of next-generation security systems with real-time responsiveness and intelligent decision-making capabilities has become a pivotal research focus.

DAS technology enables seamless monitoring across tens of kilometers of perimeter, accurately detecting intrusion behaviors (e.g., climbing, digging, and cutting) while significantly reducing false-alarm rates. A representative perimeter security system is shown in Figure 10. Glaser et al. (2022) [121] established a vibration signature database for polar environments and successfully identified simulated polar bear movements at Arctic research stations. Sun et al. (2024) [122] achieved a classification accuracy of 99.61% for airport security applications by integrating time-frequency transformations with lightweight neural architectures. Zhang et al. (2024) [123] proposed a multi-scale feature fusion technique that enables the precise classification of four intrusion scenarios using limited training data. Subsequent advances include breakthroughs in long-range localization (Sun et al., 2024 [124]), noise robustness (Xu et al., 2025 [125]), multi-signal perception (Tomasov et al., 2025 [126]), and real-time system implementation (Hu et al., 2025 [127]). Notably, Sun et al. (2025) [128] introduced a multivariate variational mode decomposition (MVMD) framework, achieving ultra-high localization precision over 101 km of fiber optic sensing, thereby establishing a new benchmark for extended-range monitoring.

By integrating digital twin modeling and edge computing, DAS systems further enable real-time intrusion detection and localization, forming a closed-loop sensing–assessment–warning security architecture. This paradigm shift transforms conventional passive alarms into proactive early warning systems, marking a significant advancement in intelligent perimeter protection.

### 4.3. Infrastructure Monitoring

The structural integrity of modern infrastructure, such as bridges, tunnels, pipelines, and power grids, plays a pivotal role in ensuring socioeconomic resilience. Confronted with challenges such as extreme weather events, material degradation, and escalating load demands, the development of high-precision real-time structural health monitoring (SHM) systems across the infrastructure lifecycle has emerged as a critical global research priority. As the physical backbone of urban operations, infrastructure health directly affects public safety and urban resilience.

In recent years, DAS has achieved significant breakthroughs in infrastructure monitoring. Figure 11 shows the current and prospective applications in this domain. In pipeline monitoring, fiber optic sensing cables enable the real-time detection of deformation induced by geological activity or construction, precise localization of corrosion and weld-cracking risks, and identification of unauthorized excavation via acoustic spectral analysis. Hussels et al. (2019) [129] optimized the sensor configuration and high-frequency data acquisition to achieve accurate localization of transient pipeline events. Zhan et al. (2025) [130] developed DPR-Net, a neural network integrated with DAS, substantially improving coverage and noise immunity in natural gas pipeline leakage detection. Additionally, Li et al. (2025) [131] successfully deployed DAS to monitor wire breakage and concrete cracking in prestressed concrete cylinder pipes (PCCPs).

In bridge monitoring, DAS technology demonstrates high sensitivity in detecting micro-strains caused by vehicle overloads and crack propagation due to foundation settlement. Rodet et al. (2025) [133] utilized urban dark fibers to assess structural health by tracking the vibration frequencies, damping ratios, and modal parameters, validating DAS feasibility for bridge applications. Moreover, the inherent advantages of DAS, including electromagnetic interference resistance and corrosion resilience, make it particularly suitable for detecting tunnel lining leakage and fatigue in wind turbine blades. Gao et al. (2024) [134] demonstrated real-time vibration monitoring in subway tunnels using optical fibers, offering critical technical support for metro safety. Xu et al. (2025) [135] further advanced this field by integrating Φ-OTDR and OFDR techniques, enabling high-precision synchronous strain monitoring of offshore wind turbine towers under harsh environmental conditions.

### 4.4. Earthquake Monitoring

Earthquakes are among the most devastating natural disasters, necessitating advanced monitoring for early warning and disaster mitigation. Conventional monitoring relies on point-based instruments, such as seismometers and strong motion sensors, which provide precise recordings of seismic waves but face limitations due to high deployment costs, sparse spatial coverage, and data transmission delays. These constraints limit real-time, large-scale dynamic monitoring capabilities, particularly in geologically complex regions and remote fields, where significant monitoring gaps persist.

DAS technology overcomes these challenges by enabling high-sensitivity detection of seismic waves and facilitating wide-field real-time monitoring. In addition, DAS can reconstruct subsurface structures through passive-source imaging or active-source inversion, making it effective for monitoring fault zones, volcanic activity, and aftershock sequences. As illustrated in Figure 12, repurposing submarine fiber optic cables to establish a global monitoring network substantially addressed the historical lack of oceanic seismic data. Azzola et al. (2022) [136] demonstrated via real-time seismic monitoring at a geothermal power plant that DAS outperformed traditional seismometers in detecting micro-seisms in noisy environments, thereby delivering superior accuracy and enhanced real-time performance. Shinohara et al. (2022) [137] further validated this by utilizing submarine cables along the coast of Japan, achieving a meter-scale spatial sampling resolution and a monitoring range exceeding 50 km. Despite its advantages, Van et al. (2025) [138] noted that DAS may experience data saturation during high-magnitude earthquakes, compromising detection accuracy and near-source monitoring efficacy.

Recent algorithmic advancements have significantly improved the performance of DAS. Biondi et al. (2024) [140] converted telecom fibers into a dense seismic array, integrating machine learning to achieve earthquake onset detection and magnitude estimation within 2.5 s. Choi et al. (2024) [141] developed a transfer-learning algorithm that achieved 97.9% accuracy in phase detection for diverse micro-seismic events. Roshdy et al. (2025) [142] conducted high-resolution seismic monitoring of near-surface structures in reservoir areas and achieved robust detection accuracy, even in complex geological settings. These innovations provide critical technical support for geological hazard assessments, urban structural health monitoring, and environmental seismic surveillance.

### 4.5. Oil Production Surveillance

DAS has emerged as a transformative technology in oil production surveillance, offering real-time monitoring of hydraulic fracturing, well integrity, flow profiling, and reservoir management. By converting fiber optic cables into dense acoustic sensor arrays, DAS provides unprecedented insights into downhole dynamics, enhancing operational efficiency and reducing risks. Figure 13 illustrates the deployment architecture of a state-of-the-art subsea permanent reservoir monitoring system (OptoSeis™), demonstrating how fiber optic sensing networks integrate with offshore production infrastructure for continuous reservoir surveillance.

Recent advances in cointerpretation techniques integrating DAS with distributed temperature sensing (DTS) have enabled more accurate multiphase flow characterization. For instance, Bukhamsin et al. (2016) [144] found that joint analysis of DAS-derived speed-of-sound measurements and DTS-based Joule–Thomson effects allows for the precise determination of in situ phase fractions in two- and three-phase flows, outperforming standalone interpretations. Field applications in Middle Eastern smart wells demonstrate close alignment with surface flowmeter data, validating the method’s reliability for inflow profiling. In Oman, a breakthrough application demonstrated how cemented fiber optic cables behind production casing enabled single-phase production profiling. DAS data processed through frequency-band filtering allowed for flow rate calibration at perforation intervals, whereas DTS provided complementary data with deeper investigation depth (Panhuis et al., 2021) [145]. This dual-technology approach represents a significant advancement in conventional oil producer surveillance.

In hydraulic fracturing diagnostics, DAS has transformed fracture monitoring through low-frequency strain signal analysis. Horizontal good deployments capture fracture dynamics including opening/closing events and stress shadow effects, whereas vertical wells delineate fracture geometry (Jin et al., 2017) [146]. This approach complements traditional micro-seismic monitoring by resolving near-wellbore fracture dimensions and interwell interactions. Quantitative flow-rate distribution during multistage fracturing is achieved through acoustic energy attribute analysis, with DAS-derived metrics showing strong agreement with DTS interpretations (Pakhotina et al., 2020) [147]. Such integrated diagnostics optimize perforation cluster design and fluid diversion strategies.

For injection well monitoring, advanced signal processing addresses DAS limitations in injection well applications. Wavefront energy compensation and hybrid denoising algorithms enhance CO_2_ injection profiling accuracy (Gan et al., 2024) [148]. Similarly, adaptive workflows combining detrending, spectral smoothing, and wavelet denoising mitigate baseline drift in gas wells (Deng et al., 2025) [149]. These methods enable both rapid on-site qualitative assessment and post-job quantitative analysis, supporting adaptive reservoir management.

## 5. Challenge

### 5.1. Data Scarcity

DAS data acquisition requires highly sensitive fiber-optic sensing equipment and specialized demodulation devices. Practical deployment often encounters logistical hurdles, including complex administrative approvals for fiber installation. Furthermore, the massive data streams generated impose stringent demands on the storage infrastructure, computational resources, and long-term maintenance costs. A critical bottleneck in advancing DAS research is the absence of authoritative and publicly available benchmark datasets, which severely hampers reproducibility and innovation. Table 4 summarizes existing open-access DAS datasets. DAS datasets predominantly consist of time-series strain or vibration measurements with spatial–temporal encoding. Typical sampling rates range from 50 Hz (Stanford arrays [150]) to 1 kHz (Fairbanks [151]), generating 1–10 TB/month for continuous monitoring. Advanced systems employ time-frequency representations or spatial–temporal matrices for event classification [152,153]. The DAShip dataset [153], for instance, contains 55,875 ship passage samples as 1 km × 1 min time–space images, whereas traffic monitoring datasets [154] use 512 × 512 pixel spectrograms derived from 500 Hz raw data.

DAS systems leverage high sampling rates and dense spatial sampling to deliver precise dynamic strain measurements. However, raw signals are inherently susceptible to environmental perturbations and multi-source noise interference, compromising data reliability in weak-signal applications such as micro-seismic monitoring. Recent breakthroughs in deep-learning-based denoising have reduced dependency on labeled data through adaptive feature extraction. Zhao et al. (2023) [156] proposed a self-supervised Sample2Sample framework that utilizes random sampling to generate training pairs for suppressing random noise in vertical seismic profiling (VSP) data. Ma et al. (2024) [157] developed a dual-branch blind-spot network (BSV) that integrates blind-spot mapping to enhance DAS-VSP data quality. Lapins et al. (2024) [158] introduced the weakly supervised DAS-N2N method, trained on noise pairs, to improve the signal-to-noise ratios of micro-seismic events. Huang et al. (2024) [159] designed a supervised DCRCDNet, an encoder–decoder architecture for complex noise suppression. Saad et al. (2024) [160] implemented an unsupervised framework combining bandpass filtering with CWT-based iterative signal reconstruction. Shi et al. (2025) [161] employed masked autoencoders (MAE) to denoise offshore DAS data and enhance seismic monitoring precision.

To address the scarcity of anomalous event samples, data augmentation techniques expand feature-space diversity to improve model robustness. Zhao et al. (2022) [162] combined GANs and CNNs to mitigate overfitting in small-sample scenarios. Shang et al. (2021) [163] applied k-SMOTE and DCGAN to resolve class-imbalance-induced accuracy degradation. Zhang et al. (2023) [164] simulated low-SNR and signal-shift conditions to enhance generalization. Shi et al. (2025) [165] proposed TSR-VAEGAN, which leverages background synthesis for sample-limited scenarios.

Deep generative models, particularly GANs, have achieved breakthroughs in synthesizing DAS data by learning probability distributions to produce high-fidelity strain field simulations. Shiloh et al. (2018) [166] pioneered GANs for training data generation and demonstrated improved classification via simulated data transformations. Their follow-up study (2019) [91] confirmed that synthetic labeled datasets enhanced ANN generalization in event detection.

To mitigate storage and real-time processing bottlenecks caused by massive DAS data volumes, compression techniques employ feature extraction and dimensionality reduction to preserve critical event information (e.g., frequency-domain anomalies in pipeline leaks or time-domain disturbances in perimeter intrusions). Dong et al. (2022) [167] devised a two-stage lossless compression method. Chen et al. (2025) [168] achieved 50× compression ratios using vision transformers, alleviating TB-scale seismic data burdens.

Although DAS technology grapples with data scarcity and processing bottlenecks, innovations in denoising, augmentation, and compression show transformative potential. Priority research should focus on constructing high-quality benchmark datasets, optimizing lightweight algorithms, and advancing multimodal fusion sensing architectures.

### 5.2. Complex Environmental Interference

DAS systems have demonstrated exceptional performance in event perception tasks. However, their practical deployment remains constrained by elevated false alarm rates induced by environmental interference, undermining system accuracy and operational utility.

A critical challenge is the missed detection of low-frequency vibrations in complex environments. Urban traffic and construction activities generate broadband noise, whereas natural disturbances (e.g., wind and rain) often exhibit time-frequency overlap with target event signals, rendering conventional threshold-based detection methods prone to false positives. Current research primarily focuses on SNR enhancement. For instance, Li et al. (2023) [169] proposed a phase error correction method based on trend prediction to improve the SNR of DAS systems. However, the heterogeneous nature of multi-modal noise limits the adaptability of the single-channel denoising algorithms. Recent studies have begun to explore multi-sensor data fusion techniques to more effectively discriminate between environmental noise and genuine events.

The rapid advancement of autonomous mobile robots (e.g., unmanned aerial vehicles (UAVs) and unmanned ground vehicles (UGVs)) has introduced new challenges in intrusion detection. Their highly adaptable morphology and behavior render conventional DAS-based perception methods ineffective, whereas target diversity further complicates feature extraction. Representative autonomous mobile robots are shown in Figure 14. To address these challenges, next-generation intrusion detection systems must innovate target feature recognition, environmental adaptability, and data processing capabilities to ensure security efficacy. Angelov et al. (2023) [170] developed a modular, sensor-fusion-based perimeter security system for UGVs and achieved notable improvements in detection accuracy. Chen et al. (2024) [171] combined the generalized cross-correlation time difference of arrival (GCC-TDOA) algorithm with least-squares optimization to achieve sub-meter-level localization of low-altitude UAVs in non-line-of-sight (NLOS) scenarios, making it suitable for critical security applications, such as airports. These studies demonstrated that multi-modal sensing and deep learning-based approaches can significantly enhance autonomous robot recognition while improving system robustness in complex environments.

To mitigate the challenges posed by environmental interference, future DAS systems should prioritize multi-modal sensor fusion and adaptive signal processing, and integrate deep learning techniques to establish intelligent, real-time, and high-speed analytical frameworks. These advancements will enhance noise immunity, reduce false alarms, and drive the DAS systems toward greater reliability and intelligence.

## 6. Conclusions

DAS has emerged as a pivotal technology in modern acoustic surveillance due to its advantages in long-range, high-precision, and real-time monitoring. This paper systematically reviews recent advancements in event detection techniques for DAS systems. First, the fundamental principles, system architecture, and performance metrics of DAS technology are briefly outlined. Next, we comprehensively categorized event detection algorithms in DAS, encompassing traditional ML, DL, and hybrid methodologies. Subsequently, we provide an in-depth analysis of DAS applications across diverse domains, including urban road and railway monitoring, perimeter security intrusion detection, infrastructure health assessment, and seismic monitoring. Finally, we critically evaluate the key challenges hindering further advancements in DAS technology and propose future research directions.

In algorithm design, traditional ML methods retain their advantages in resource-constrained scenarios owing to their computational efficiency. DL models achieve state-of-the-art detection accuracy through spatiotemporal–frequency joint feature extraction. Hybrid architectures, which combine deep neural networks with optimized shallow classifiers, further strike an optimal balance between computational efficiency and precision, facilitating robust DAS deployment in complex environments.

In transportation monitoring, DAS enhances vehicle detection, rail defect diagnostics, and early risk warning systems. Perimeter security enables high-precision, long-range, all-weather intrusion detection. Infrastructure assessment benefits from DAS-enabled structural health monitoring of pipelines, bridges, and tunnels. In seismology, DAS improves micro-seismic detection sensitivity and early warning timeliness. Emerging applications such as wastewater discharge monitoring and polar wildlife tracking further underscore its versatility.

Despite this promise, several critical challenges remain. The primary obstacle lies in the data scarcity dilemma: publicly available benchmark datasets remain limited, whereas real-world scenarios exhibit a severe imbalance in anomalous event samples. This necessitates advances in few-shot generation techniques to mitigate the data insufficiency. Another significant challenge stems from environmental interference, where strong noise contamination results in persistently high false alarm rates. Addressing this issue requires multi-modal sensor fusion and adaptive signal processing technologies to enhance the SNR.

Future research should focus on developing collaborative learning frameworks that enable knowledge sharing across distributed DAS deployments without compromising data privacy. Federated learning approaches could prove particularly valuable for cross-border infrastructure monitoring or multi-operator urban sensing networks, where centralized data collection is impractical. Another promising direction involves the integration of DAS with complementary sensing modalities such as LiDAR or satellite imagery. By combining acoustic signatures with visual or thermal data through advanced fusion algorithms, systems could achieve unprecedented robustness against environmental noise and improved discrimination of complex events.

The adaptation of large pre-trained models to DAS applications presents another exciting opportunity. Foundation models developed for related domains like seismic analysis or computer vision could be fine-tuned for fiber optic sensing tasks, potentially overcoming current limitations in labeled training data. Meanwhile, the development of physics-informed neural networks that incorporate wave propagation principles could enhance model interpretability and performance in extreme environments. On the hardware front, innovations in edge computing and neuromorphic processors may enable new generations of energy-efficient, low-latency DAS systems capable of real-time processing for critical applications like earthquake early warning.

Ultimately, the convergence of advanced optical sensing, distributed computing, and machine learning promises to unlock new capabilities for DAS systems. By addressing current limitations while exploring these emerging directions, researchers can position DAS as a cornerstone technology for next-generation smart infrastructure and environmental monitoring systems. The coming years will likely witness transformative applications as DAS evolves from a specialized monitoring tool to a ubiquitous sensing platform supporting critical decision-making across industries and government agencies.

## Figures and Tables

**Figure 1 sensors-25-05052-f001:**
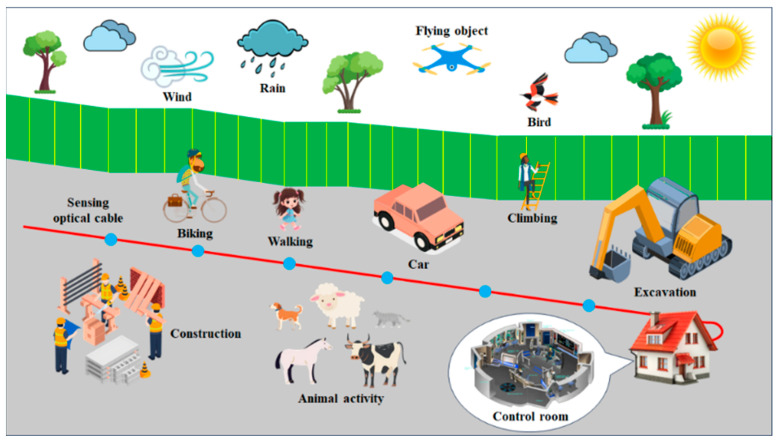
A schematic diagram of target types for event perception in DAS.

**Figure 2 sensors-25-05052-f002:**
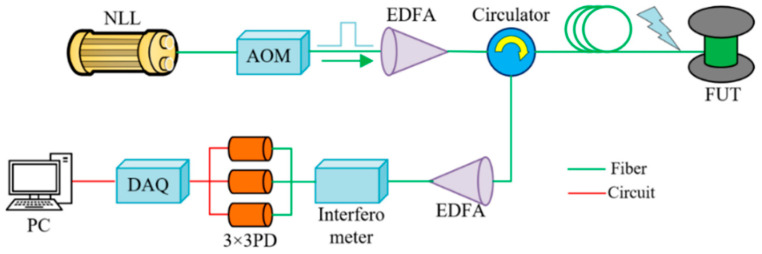
Distributed acoustic sensing system.

**Figure 3 sensors-25-05052-f003:**
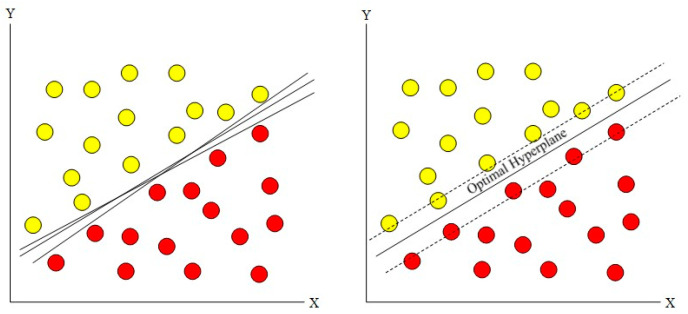
Drawing of a linear optimal hyperplane generated by the SVM model.

**Figure 4 sensors-25-05052-f004:**
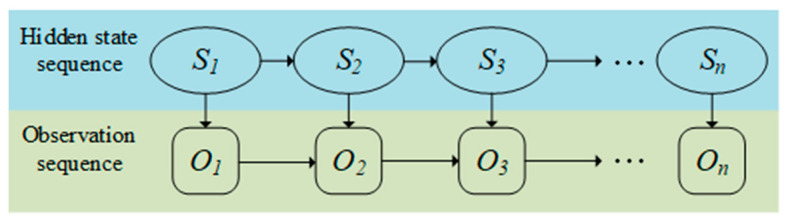
Hidden Markov model sequence.

**Figure 5 sensors-25-05052-f005:**
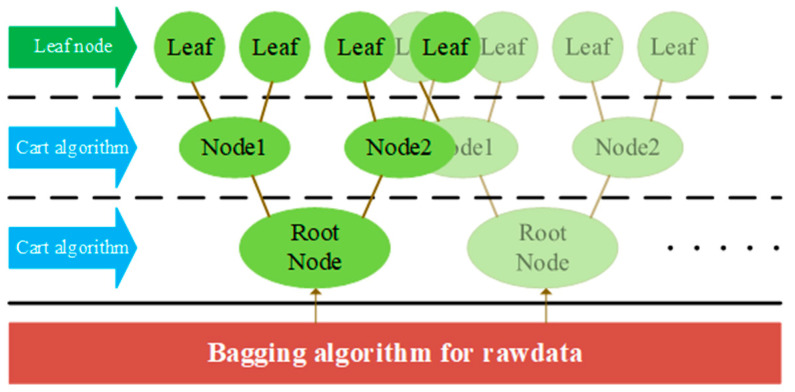
Schematic diagram of the random forest algorithm.

**Figure 6 sensors-25-05052-f006:**
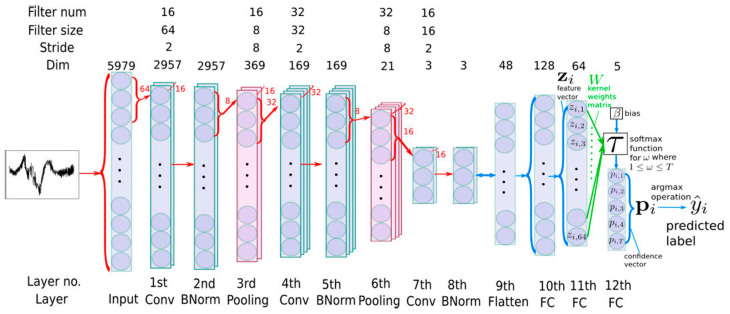
1D-CNN architecture for DAS signal classification (Ref. [67], Figure 5). Conv: Convolutional layers. BNorm: Batch normalization. Pooling: Pooling layers. Flatten: Flatten layer. FC: Fully connected layers.

**Figure 7 sensors-25-05052-f007:**
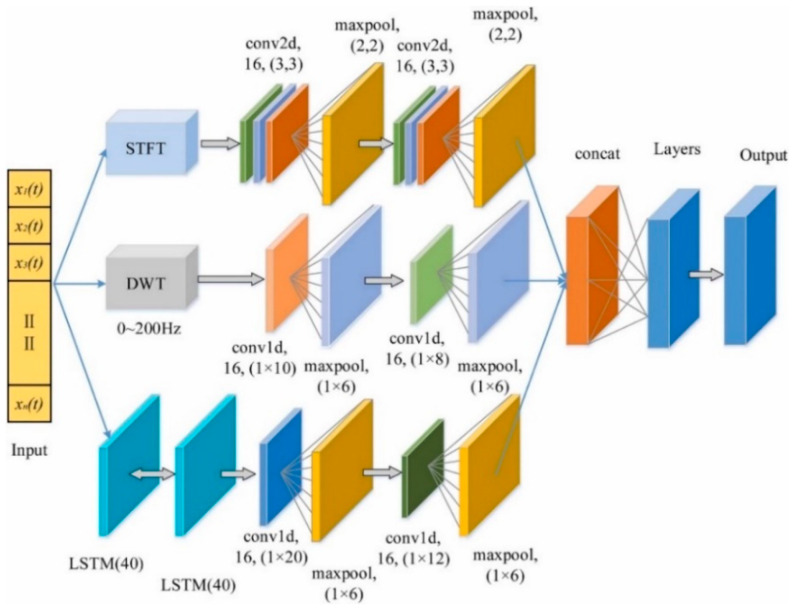
Structure of LSTM-CNN (Ref. [92], Figure 8).

**Figure 8 sensors-25-05052-f008:**
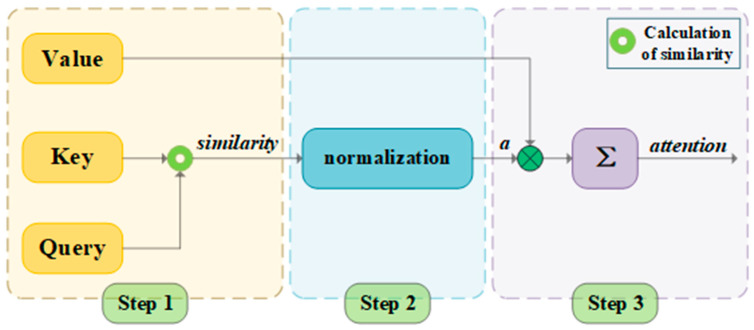
A general model of the attention mechanism. The system transforms input features into query (Q), key (K), and value (V) matrices through learned linear projections.

**Figure 9 sensors-25-05052-f009:**
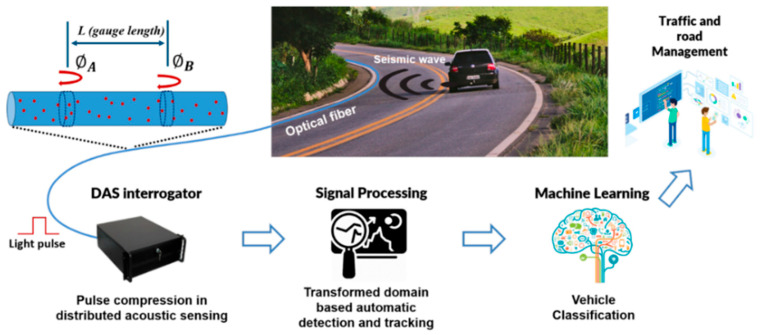
Schematic diagram of DAS-based traffic monitoring system (Ref. [115], Figure 1).

**Figure 10 sensors-25-05052-f010:**
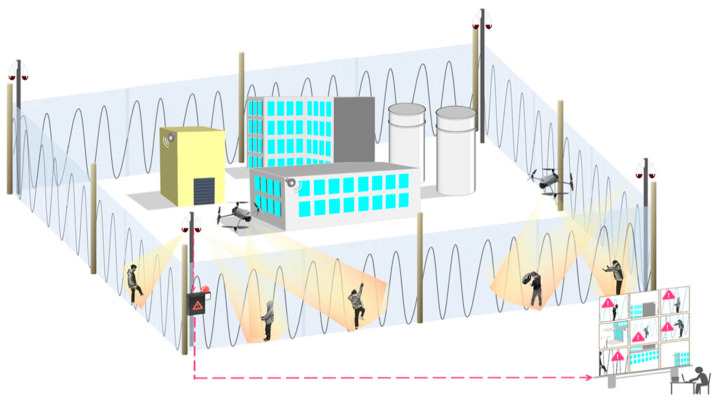
Fiber optic and hardware-linked perimeter security system (Ref. [122], Figure 1). (1) Sensing fiber deployed along fences with optimized bending radius; (2) 3 × 3 coupler interferometer for phase demodulation; (3) FPGA-accelerated edge computing unit; and (4) multi-tier alarm classification. Pink arrows indicate signal flow direction.

**Figure 11 sensors-25-05052-f011:**
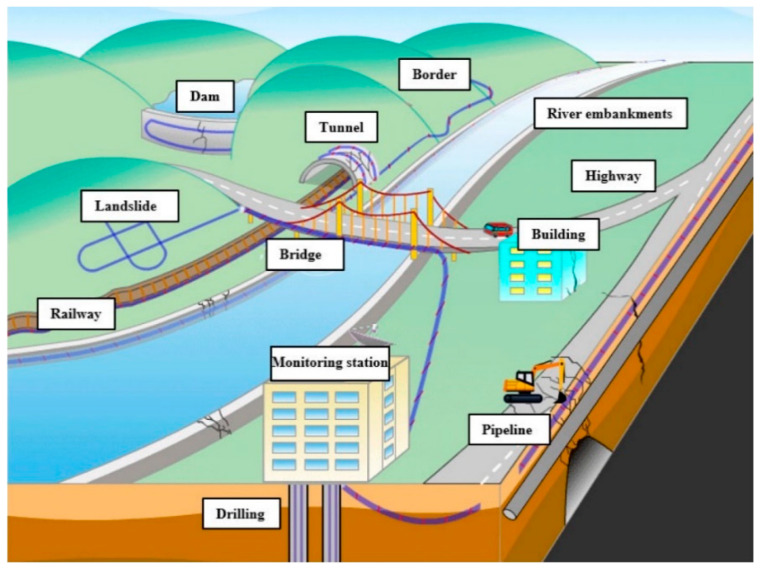
Current and potential application scenarios of DAS for monitoring infrastructure and geological hazards (Ref. [132], Figure 1).

**Figure 12 sensors-25-05052-f012:**
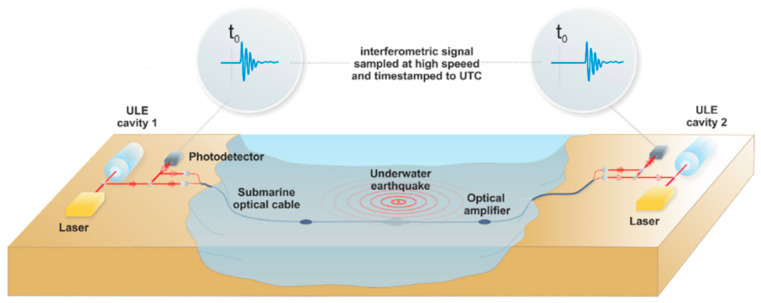
Diagram of the principle of epicenter location (Ref. [139], Figure 5).

**Figure 13 sensors-25-05052-f013:**
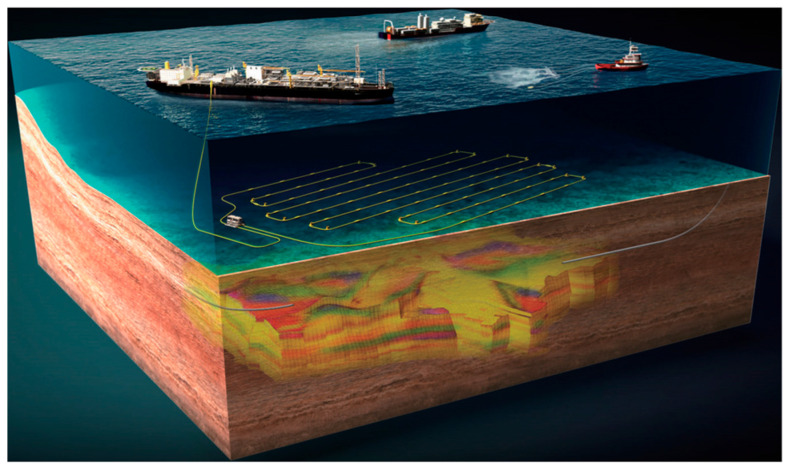
OptoSeis^TM^ subsea permanent reservoir monitoring system deployment and operation diagram (Ref. [143], Figure 15).

**Figure 14 sensors-25-05052-f014:**
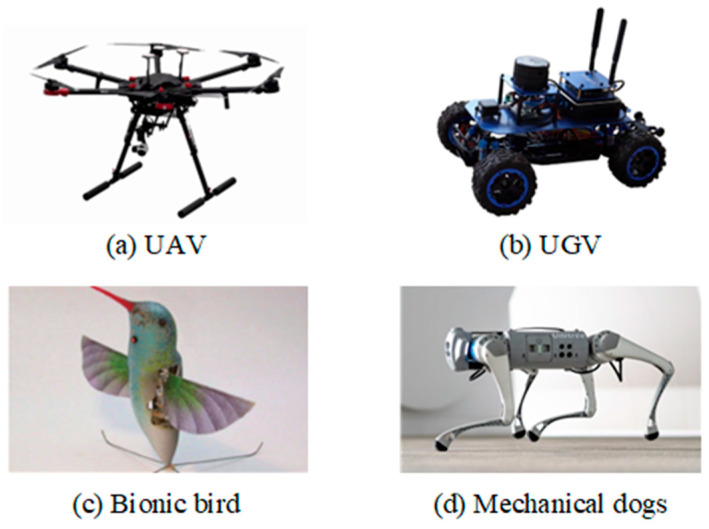
Some typical autonomous mobile robots.

**Table 1 sensors-25-05052-t001:** Summary of DAS event perception technology based on machine learning.

Reference	Year	Method	Accuracy	Application Field
[49]	2020	SVM + Kalman Filter	>98%	Railway Monitoring
[50]	2020	LSVM + VMD	75.2%/79.5%	Intrusion Detection
[51]	2021	SVM + Wavelet Denoising + Chebyshev Filter	>97%	Railway Monitoring
[52]	2024	SVM + CV	99.62%	Pipeline Monitoring
[53]	2019	HMM + Knowledge Mining	98.2%	Pipeline Monitoring
[54]	2024	HMM + Waveform Segmentation	97.30%	Railway Safety
[55]	2019	RF	96.58%/99.32%	-
[56]	2021	RF + EMD	92.31%	Tunnel Monitoring
[57]	2024	RF + Matched Filter + Root Mean Square	98%	Traffic Monitoring
[58]	2022	IF	>90%	Infrastructure Monitoring
[59]	2024	Logistic Regression	93.4%	Pipeline Monitoring
[60]	2017	MLP + GMM	54.92%/69.7%	Pipeline Monitoring
[61]	2020	DBSCAN Clustering	-	Tunnel Monitoring
[62]	2025	Principal Eigenvalue Analysis + FastICA	-	-

**Table 2 sensors-25-05052-t002:** Summary of DAS event perception technology based on deep learning.

Reference	Year	Method	Accuracy	Application Field
[63]	2021	CNN	98.04%	Railway Safety
[64]	2021	IP-CNN	88.2%	-
[65]	2021	FC-ANN + CNN + RNN	96.94%/93.86%	Earthquake Monitoring
[66]	2022	CNN + Greedy Algorithm	97.91%	Railway Safety
[67]	2023	1-D CNN	>94%	Traffic Monitoring
[68]	2024	TFF-CNN	99.30%	Intrusion Detection
[69]	2024	1-D MFCNN + 1-D MFEWnet	99.6%	Perimeter Security
[70]	2024	3-D ACNN	99.33%	-
[71]	2023	CEEMDAN-Permutation Entropy + RBF	88.15%	Pipeline Monitoring
[72]	2024	STNet + SW	96.9%	Intrusion Detection
[73]	2024	Multi-task Learning + CNN	>96%	Perimeter Security
[74]	2024	Faster R-CNN	98.85%	Pipeline Monitoring
[75]	2020	YOLOv3	>80%	Earthquake Monitoring
[76]	2025	YOLOv5-Break	97.72%	Pipeline Monitoring
[77]	2024	YOLOv7 + CBAM	99.7%	-
[78]	2024	YOLOv8 + CBAM	97.78%	Perimeter Security
[79]	2023	YOLOX	100%	Infrastructure Monitoring
[80]	2020	ConvLSTM + CNN	90%	Railway Safety
[81]	2020	ConvLSTM + CNN	85.6%	Railway Safety
[82]	2023	CNN + LSTM	93.87%	-
[83]	2019	CLDNN	>97%	Pipeline Monitoring
[84]	2024	1D CNN + Bi-LSTM	>94.5%	-
[85]	2024	CNN-LSTM-SW	97%	Railway Safety
[86]	2024	LSTM + GRU	>93%	Railway Monitoring
[87]	2024	ConvLSTM	72.8%	Human Flow Monitoring
[88]	2023	MS-CNN	95.43%	Threat Event Detection
[89]	2024	DSAD + DSAD-VAE	100%	Rail Safety
[90]	2024	CNN-LSTM-Self-Attention	96.25%	Intrusion Detection
[91]	2019	SimGAN	80.2%	-
[92]	2022	DAE	-	Traffic Management
[93]	2023	NAM-MAE	96.6134%	Peripheral Security
[94]	2024	CNN + VQ-VAE	95%	Natural Disaster Monitoring

**Table 3 sensors-25-05052-t003:** Summary of DAS event perception technology based on the mixture of machine learning and deep learning.

Reference	Year	Method	Accuracy	Application Field
[95]	2019	CNN + K-means	<90%	Pipeline Monitoring
[96]	2020	CNN + SVM	94.17%	-
[97]	2021	CNN + SVM + RF	>99%	Pipeline Monitoring
[98]	2022	Decision Tree + BP Neural Network	97.6%	Perimeter Security
[99]	2023	Convolutional AE + Clustering	91.5%	Railway Safety
[100]	2023	Meta-learning + Wide ResNet	97.65%/98.80%/98.85%	Infrastructure Monitoring
[101]	2020	MSCNN + Prototype Learning	84.67%	Perimeter Security
[102]	2025	YOLOv7 tiny + Greedy Algorithm	81.8%/60.4%	Perimeter Security
[103]	2023	U-Net + FBE	-	Pipeline Monitoring

**Table 4 sensors-25-05052-t004:** Publicly available DAS dataset.

Dataset Name	Year	Application Field	Events (Quantity)	Data Type	Key Features
Railway Track Performance [114]	2020	Railway monitoring	Train loads, track strain, track displacement, and bending (~5 GB)	Time-series strain	10 km fiber, vibration events
PubDAS [150]	2023	Geosciences	Urban centers, underground mining areas, submarine earthquakes, anthropogenic noise, and natural phenomena (~90 TB)	Seismic waveforms	8 global field experiments
Brady Hot Springs [151]	2018	Geothermal monitoring	Earthquakes: 4 + main (~90 TB)	Temperature/Strain	15 day continuous DAS + DTS
Φ-OTDR Events [152]	2023	Event classification	Background noise (3094), excavation (2512), tapping (2530), watering (2298), shaking (2728), and walking (2450) (15 GB)	Time-space matrices	Labeled human/mechanical activities
DAShip [153]	2025	Maritime security	Ship passages (55,875) and ship instances (18,625)	Time-space images	3 TB total, AIS-correlated
Intelligent Traffic DAS [154]	2024	Traffic monitoring	Vehicle patterns and noise interference (200 sample pairs)	Spectrograms	Paired raw/processed data
Subsea Cable [155]	2019	Submarine cables	Cable impacts and cyclic cable loading (45 GB)	Vibration signals	131 km fiber span

## Data Availability

Not applicable.

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
