# Peer review of "Research Progress of Event Intelligent Perception Based on DAS"

_sensors, 2025, doi:10.3390/s25165052_

Round 1
Reviewer 1 Report
Comments and Suggestions for Authors
This review systematically summarizes the research progress of Distributed Acoustic Sensing (DAS) in the field of intelligent event perception. It covers the fundamental principles of DAS, system architecture, performance metrics, event recognition methods (including traditional machine learning, deep learning, and hybrid models), representative applications (such as traffic monitoring, security defense, infrastructure surveillance, and seismic early warning), as well as current challenges (including data scarcity and environmental interference). However, the manuscript still contains numerous issues that require further revision and improvement:
- Multiple instances of incorrect figure references such as “Figure Figure 3” and “Figure Figure 6” appear throughout the text and should be consistently corrected.
- Some illustrations (e.g., Figures 6, 8, and 10) lack in-depth explanations in their captions. It is recommended to supplement these figures with more detailed descriptions.
- The descriptions of datasets in Table 4 are somewhat fragmented and require a concise summary to enhance clarity.
- Although Tables 1–3 present algorithm performance metrics, they do not provide a comparative analysis of which model types are better suited for specific conditions such as low-sample environments, high-noise settings, or real-time processing. It is recommended to add a subsection that summarizes the applicability of different models under various scenarios.
- In Chapter 6, the discussion of future directions should be expanded to include emerging topics such as federated learning, multi-modal joint modeling, and the transferability of large pre-trained models to DAS systems, in order to enhance the paper’s forward-looking perspective.
- Several schematic diagrams (e.g., Figures 6 and 8) contain numerous abbreviations. It is advisable to explain these terms either in the figure captions or in the main text.
- Equation (6) is incorrect. It is recommended to revise it with reference to the following literature:
Lindsey, N. J., Rademacher, H., & Ajo‐Franklin, J. B. (2020). On the broadband instrument response of fiber‐optic DAS arrays. Journal of Geophysical Research: Solid Earth, 125(2), e2019JB018145.
- DAS monitoring has already been widely applied in oil production surveillance, which is not addressed in the current manuscript and should be supplemented. A recommended reference is:
Jin, G., & Roy, B. (2017). Hydraulic-fracture geometry characterization using low-frequency DAS signal. The Leading Edge, 36(12), 975–980.
Bukhamsin A, Horne R. Cointerpretation of distributed acoustic and temperature sensing for improved smart well inflow profiling[C]//SPE Western Regional Meeting. SPE, 2016: SPE-180465-MS.
Deng R, Gan L, Shi Y, et al. Data Processing Method of Das Logging[C]//SPWLA Annual Logging Symposium. SPWLA, 2025: D041S013R004.
Pakhotina I, Sakaida S, Zhu D, et al. Diagnosing multistage fracture treatments with distributed fiber-optic sensors[J]. SPE Production & Operations, 2020, 35(04): 0852-0864.
Gan L, Dang L, Wang D, et al. Research on the processing and interpretation methods of distributed fiber optic vibration signal logging injection profiles[J]. Geoenergy Science and Engineering, 2024, 239: 212980.
Author Response
We sincerely appreciate the reviewer’s thorough and constructive feedback, which has significantly improved the quality of our manuscript. Below, we provide point-by-point responses to all comments and detail the revisions made in the revised manuscript. All changes are highlighted in red in the tracked-changes version.
Comment 1: Multiple instances of incorrect figure references such as “Figure Figure 3” and “Figure Figure 6” appear throughout the text and should be consistently corrected.
Response 1: We apologize for these typographical errors. All incorrect figure references (e.g., duplicate "Figure" labels) have been corrected in the revised manuscript (see Pages 7, 10).
Comment 2: Some illustrations (e.g., Figures 6, 8, and 10) lack in-depth explanations in their captions. It is recommended to supplement these figures with more detailed descriptions.
Response 2: Agree. We have expanded the captions for Figures 6, 8, and 10 to include detailed descriptions of the abbreviations, methodologies, and key takeaways (see Pages 10, 12, 17).
Comment 3: The descriptions of datasets in Table 4 are somewhat fragmented and require a concise summary to enhance clarity.
Response 3: Agree. We have restructured Table 4 by adding descriptions of data types and key characteristics for each dataset, along with a summary paragraph (Page 21) highlighting the commonalities and limitations across datasets.
Comment 4: Although Tables 1–3 present algorithm performance metrics, they do not provide a comparative analysis of which model types are better suited for specific conditions such as low-sample environments, high-noise settings, or real-time processing. It is recommended to add a subsection that summarizes the applicability of different models under various scenarios.
Response 4: Agree. We have added a new subsection titled "Comparative Analysis of Model Suitability Across Operational Scenarios" in Section 5 of Chapter 3 (Pages 14–15). Key additions: A bullet-point summary of model strengths (e.g., Deep learning excels in high-noise settings; Hybrid models balance real-time needs). Citations to support these comparisons.
Comment 5: In Chapter 6, the discussion of future directions should be expanded to include emerging topics such as federated learning, multi-modal joint modeling, and the transferability of large pre-trained models to DAS systems, in order to enhance the paper’s forward-looking perspective.
Response 5: Agree. We have added three new paragraphs (Page 24) discussing emerging trends, including: Federated learning for privacy-preserving DAS applications, Multi-modal joint modeling (e.g., integrating DAS with LiDAR data). Potential of large pre-trained models.
Comment 6: Several schematic diagrams (e.g., Figures 6 and 8) contain numerous abbreviations. It is advisable to explain these terms either in the figure captions or in the main text.
Response 6: Agree. All acronyms appearing in Figures 6 and 8 have been explicitly defined in their captions (Pages 10 and 12, respectively), with full terminology provided for clarity.
Comment 7: Equation (6) is incorrect.
Response 7: Agree. We have revised Equation (6) (Page 5) based on the referenced literature (Lindsey et al., 2020. On the broadband instrument response of fiber‐optic DAS arrays.).
Comment 8: DAS applications in oil production surveillance are missing.
Response 8: Agree. We have added a new subsection (Page 19–20) titled “Oil Production Surveillance”, synthesizing both suggested references and our comprehensive literature review. The content systematically addresses: (1) Distributed acoustic sensing for hydraulic fracturing diagnostics; (2) Advanced inflow profiling in intelligent well completions; and (3) Case studies on DAS applications in oilfield monitoring.
Reviewer 2 Report
Comments and Suggestions for Authors
The paper provides a comprehensive review of the intelligent event detection methods based on Distributed Acoustic Sensing. This paper is technically sound, well-researched, and relevant to current developments in DAS and Intelligent Event Perception Techniques.
I consider a relevant and very complete review for the mentioned areas of research. In particular, the given focus on Machine learning and deep learning techniques for event classification is quite interesting and important for the scientific community. The presentation of practical applications and challenges in real-world implementation are very insightful.
Apart from English revision, I consider the work in highly suitable for publication.
The paper lacks information about dataset sizes and/or input data types (waveform, spectrogram, etc.). This should be addressed in the paper.
Author Response
We sincerely appreciate the reviewer’s thorough and constructive feedback, which has significantly improved the quality of our manuscript. Below, we provide point-by-point responses to all comments and detail the revisions made in the revised manuscript. All changes are highlighted in red in the tracked-changes version.
Comment 1: The paper lacks information about dataset sizes and/or input data types (waveform, spectrogram, etc.). This should be addressed in the paper.
Response 1: We acknowledge this oversight and have now included a dedicated section (Page 21, Table 4, Section 5.1) summarizing publicly available DAS datasets, their sizes, data types (e.g., time-series strain, spectrograms), and key features. This addition clarifies the diversity and scale of input data used in DAS event perception.
Comment 2: The paper is technically sound and well-researched but requires English revision.
Response 2: We used methods such as Grammarly to revise the grammar, clarity, and coherence of the manuscript, with particular attention to: (1) Technical terminology consistency; (2) Sentence structure simplification; and (3) Proofreading by a native English speaker to ensure fluency.